# Analysis of University Student Motivation in Cross-Border Contexts

**DOI:** 10.3390/ijerph20115924

**Published:** 2023-05-23

**Authors:** Lionel Sánchez-Bolívar, Silvia Navarro-Prado, María Angustias Sánchez-Ojeda, Victoria García-Morales, Jonathan Cortés-Martín, María Isabel Tovar-Gálvez

**Affiliations:** 1Faculty of Humanities and Social Sciences, Isabel I de Castilla University, 09003 Burgos, Spain; lionel.sanchez@ui1.es; 2Department of Nursing, Faculty of Health Sciences of Melilla, University of Granada, 52017 Melilla, Spain; silnado@ugr.es (S.N.-P.); maso@ugr.es (M.A.S.-O.); 3Physiology Area, Department of Biomedicine, Biotechnology and Public Health, Faculty of Medicine, University of Cádiz, Pl. Falla, 9, 11003 Cádiz, Spain; victoria.garcia@gm.uca.es; 4Research Group CTS1068, Department of Nursing, Faculty of Health Sciences, University of Granada, 18071 Granada, Spain; 5Department of Nursing, Faculty of Health Sciences of Ceuta, University of Granada, 51001 Ceuta, Spain; matoga@ugr.es

**Keywords:** cross-border context, education, empathy, mental health, motivation, resiliency, student behaviour, self-motivation, university

## Abstract

The development of the personality of university students can determine their affinities for certain disciplines; therefore, it is important to know their specific socio-demographic and motivational profile, what motivates them to start a certain university degree and what encourages them to continue with it, which can help to adapt the teaching methodology. A total of 292 university students from the University of Granada (Ceuta and Melilla campuses) participated in this quantitative study with a descriptive, cross-sectional design, in which motivation and social skills were analysed. Among the results, it can be highlighted that the student population is mainly female, with a higher level of motivation. Sociability, communication, thinking (optimistic or pessimistic), empathy and self-confidence are skills that affect university students’ motivation levels. This study highlights the importance and impact of students’ motivation on their learning and the development of their social competence, so it is essential to carry out educational interventions that promote these types of skills, especially in cross-border contexts, which can be demotivating environments.

## 1. Introduction

Motivation can be considered as the intentional and autonomous drive, force or will influencing the decision making, execution and outcome of an action performed by the individual [1] (Ryan and Deci, 2017). This definition of motivation has provided an important basis for completing cognitive processes such as planning, organising, decision making, learning and situational assessments [2]. Following this conceptual line, some authors define motivation as a “psychological variable resulting from the interaction of other cognitive, biological, social and emotional variables” that will be determinant and crucial in the explanation of certain human behaviour, such as: choices, “intensity and persistence in a specific activity” [3].

To explain this phenomenon, McClelland (1965) developed the Theory of Motive Acquisition, stating that motivation occurs when the outcome is pleasant (hedonistic view) and that there is an absence of motivation, or the opposite (demotivation) when the outcome is unpleasant [4].

Using Self-Determination Theory, and based on studies they conducted comparing intrinsic and extrinsic motives that influenced an individual’s decision making and behaviour, Deci and Ryan (2017) understood the importance of motivation (specifically intrinsic motivation) in these areas. As a complement to this theory, the authors formulated Organic Integration Theory to explain the situations in which extrinsic motivation was more important in the behaviour and choices of the individual. Using these two theories, they explain and justify the role of motivation in the decision-making process and in human behaviour, both extrinsic and intrinsic [1]. These theories establish the concept of amotivation as a situation in which the individual perceives that there is no relationship between his or her actions and the outcome of those actions [5,6,7,8,9].

When analysing the typology of the predominant motivation of students, there are five types of motivation: extrinsic, intrinsic, instrumental, integrative and transcendental.

*Extrinsic motivation*. Ryan and Deci summarize extrinsic motivation as the fact of doing something because of the psychological well-being obtained from the result [1]. Therefore, this type of motivation is conditioned by external factors [10,11]. An initially intrinsic motivation, through the application of rewards or punishments, can evolve into extrinsic motivation (externalization) as a response to the desire to obtain a reward or avoid a punishment. This externalization process starts from intrinsic motivation, and motivation moves through four levels of regulation: The first level, integrative regulation, is where the activity is incorporated into everyday actions because it already provides psychological well-being. Subsequently, regulation becomes identified—where the behaviour performed is highly valued, the individual will perform it even if he/she does not like it (for example: practicing sport for a short time but doing it because he/she knows that it has physical and psychological benefits). After this, the introjected regulation externalizes the locus. As a step towards the extrinsification of motivation, external regulation is established [1,10,12]. However, the authors establish the possibility of internalizing extrinsic motivation, gradually transforming it into intrinsic motivation, calling this process internalization [13,14,15].

*Intrinsic motivation*. Intrinsic motivation is conceived as that which is given by the action, i.e., the motivating agent is the act performed in itself; therefore, the motivation comes from within the person. This can arise from the mere fact of feeling wellbeing with the performance of the action, although it can also be produced by the repetition of the action, transforming an extrinsic motivation into an intrinsic one (by internalization of the action) [8,16,17,18,19].

*Instrumental motivation*. Someone is said to be instrumentally motivated when actions are aimed at achieving goals other than integration goals, such as passing exams, improving careers, etc. [20]. Instrumental motivation supports the desire for social recognition and economic benefits. Therefore, this motivation refers to functional reasons and utility value in achievement [21,22,23].

*Integrative motivation*. Integrative motivation can be seen as the desire or willingness to identify with a social group. Therefore, the individual identifies in the group values that he or she shares. It is also driven by the impulse to be part of it, born from the person but through an external activator [24,25].

*Transcendent motivation*. This type refers to the drive or force that moves the individual by the mere fact that the action will be beneficial to another person or persons [26]. Therefore, this motivation relates the intrinsic factor of wanting to do something for an internal interest with the psychological well-being that is obtained by verifying that it is beneficial for other individuals [27].

It should be noted that the nature and level of motivation varies across levels of preschool, primary, secondary, baccalaureate, vocational and higher education.

In early childhood, it is directly related to the behaviour of teachers and the reinforcement or punishment students receive from teachers [8,28,29,30,31].

In elementary school, motivation is associated with goal setting and self-efficacy, increased extrinsic motivation strength and decreased closeness to teachers. It is based on reward-oriented motivation and punishment avoidance [32,33,34,35].

In secondary school, gender stereotypes are reinforced, which establishes an extrinsic and integrative pattern of motivation in students due to the need to be part of the peer group, as well as fear of exclusion [36].

In university students, motivation is linked to self-regulation [37] and self-efficacy. There are considerable differences in the levels of this triad (motivation–self-regulation–self-efficacy) between Law, Physics, Philosophy, Biology and Religious Studies [38]. Amotivation and demotivation are related to tendencies towards procrastination [39], whereby a decrease in motivation would lead to an increase in procrastination and vice versa, initiating the process of amotivation (and, subsequently, demotivation), reaching the point of academic abandonment in cases of high amotivation or demotivation.

The cross-border context can have a number of influences on university students. In these environments they are likely to encounter people from different cultures and ethnic backgrounds. This can provide a valuable multicultural experience, allowing them to learn from different perspectives and expand their understanding of the world. Thus, these environments may make it difficult for university students to adapt to environments where there may be cultural barriers, added to the difficulty of moving to an environment—the university—where a certain amount of family independence is experienced. This can lead to stress and demotivation, although it can also provide opportunities for personal growth and development [2,40,41,42,43,44,45,46].

With the aim of adapting the academic methodology to university students, various analyses of their profile were carried out. In this sense, one of the main factors that concerns the research community, as far as university students are concerned, is their motivation to study the degree they are currently studying and what drives them to continue, complete and, in the future, develop it professionally.

Knowing what motivates students to make decisions or act as they do will provide teachers with guidelines to adapt, develop and optimise both the theoretical–practical methodological approach, and the didactic and institutional resources at their disposal, for the greater academic benefit of the students; therefore, this study aims to describe and analyse what type of motivation is predominant among students in the cross-border cities of Ceuta and Melilla (Spain).

## 2. Materials and Methods

This research employs a descriptive and exploratory design of a cross-sectional nature, using a relational study to analyse the degree of relationship between the variables defined.

### 2.1. Population and Sample

In order to carry out this study, a sample of 292 university students (66.8% (N = 195) women and 33.2% (N = 97) men) enrolled in different undergraduate studies at the university campuses of Ceuta (representing 78.4% (N = 229)) and Melilla (22.6% (N = 63)), both belonging to the University of Granada, was taken by means of purposive sampling, assuming a sampling error of 5%.

### 2.2. Variables and Instruments

This research study operationalizes and quantifies several variables through the use of specific instruments. These variables include demographic factors such as age, gender, campus location and religion. Additionally, the study measures academic variables, including the degree studied and the reason for enrolment, as well as social skills and motivation. These social skills are collected by an ad hoc, Likert-type scale from 1 to 10, with ten items, in which two polarised options are given—among which are the 1–10 scale—referring to different social skills. The items set are Introvert–Extrovert; Passive–Active; Reactive–Proactive; Asocial–Sociable; Reserved-Communicative; Pessimistic–Optimistic; Indifferent–Empathetic; Aggressive–Assertive; Submissive–Dominant; and Inflexible–Adaptive.

To assess motivation, the Situational Motivation Scale, adapted to the Spanish educational context by Martín-Albo et al., was used [47,48]. Using this scale, we found 16 items that assess the four dimensions of motivation mentioned above: intrinsic motivation (items 1, 5, 9, 13), identified regulation (items 4, 8, 12, 16), external regulation (items 3, 7, 11, 15) and amotivation. The instrument obtained an overall validity of α = 0.700 and a subscale validity of 0.858 for intrinsic motivation, 0.757 for identified regulation, 0.700 for external regulation and 0.774 for amotivation.

### 2.3. Procedure

The data collection and completion of questionnaires was carried out during the months of April and May 2019. One of the researchers was available at all times to advise on the completion of the questionnaires and to resolve any doubts raised by the students. After data collection, the questionnaires were organised and coded. In the process of coding and data dumping, a total of 8 incompletely questionnaires were detected, so they were discarded.

The SPSS 25.0 programme (Statistical Package for the Social Sciences) (IBM Corp.: Armonk, NY, USA) provided by the University of Granada was used to analyse the results.

After applying the normality and homoscedasticity tests, the Kolmogorov–Smirnov test was used to check that the sample followed a normal distribution. After checking that factorial techniques could be used, the results were analysed.

The first section involves a descriptive analysis of the questionnaire data, while the second section uses a one-factor ANOVA to establish the relationship between polytomous variables and Student’s t-test for dichotomous variables.

When applying the instrument, the students were informed of its voluntary and anonymous application, and they were required to indicate in the questionnaire whether they consented to participate in the study by ticking a consent box. This research followed the ethical standards of the Declaration of Helsinki and the Human Research Ethics Committee of the University of Granada, with the ethical code 2950/CEIH/2022.

## 3. Results

According to the data, the average age of the participants is 22.03 (±5.8) years, with the youngest being 18 years old and the oldest 54 years old. The socio-demographic profile of the population studied is presented in Table 1, where the majority (66.8% or N = 195) are female, and the remaining 33.2% (97 individuals) are male.

The population from the Ceuta Campus represents 78.4% (N = 229) of the population, while that of the Melilla Campus is limited to the remaining 21.6% (N = 63). As can be seen in Table 1, on the Ceuta Campus, 54.5% (N = 159) of the population is female, while 24% (N = 70) is male. On the Melilla Campus, the data are more equitable, representing 9.2% (N = 27) of men and 12.3% (N = 36) of women.

In terms of religion, the majority religion is Catholic/Christian (N = 163; 55.8%), followed by Muslim (N = 75; 25.7%).

Regarding the students’ degrees, the predominant population with a percentage of 34.6% (N = 101) is the bachelor’s degree in Nursing on both campuses. This is followed by the degree in Business Administration and Management, with 18.5% (N = 54); degree in Primary Education with 14% (N = 41); degree in Early Childhood Education with 12.3% (N = 36); degree in Social Education with 11.3% (N = 33); degree in Primary Education and Physical Activity and Sport Sciences with 6.8% (N = 20); with the last positions being occupied by the degree in Business Administration and Management with 2.1% (N = 6) and Law; and the degree in Labour Relations and Human Resources with 0.3% (N = 1).

The main motivation expressed by the students (Table 2) is “vocation” with 49.70% (N = 145); followed by “preference” (understood as the most interesting option for the students to choose) with 14.40% (N = 42); with “having more professional knowledge” in last place with 0.7% (N = 2) and other motivations such as “family expectations” and “increase of professional curriculum”.

The results generally establish a good level of social skills among the university students. In this line, it is worth noting that the most developed social skill is “adaptability” (N = 265; 90.8%), which is understood as the ability to adapt to different situations and contexts. Closely following this, we find “empathy” with 59.9% (N = 175) and “assertiveness” with 52.1% (N = 152). Special attention should also be paid to the contrast of “Optimism-Pessimism”, where although a majority of 65.4% (N = 191) consider themselves to be optimistic, 34.6% (N = 101) consider themselves to be pessimistic. Similarly, we find this contrast, with similar data, in the dyads “Introversion-Extroversion”, with 33.2% (N = 97) of introverts and 66.8% (N = 195) of extroverts; “Reactivity-Proactivity”, with the reactive population representing 31.5% (N = 92) and the proactive population 68.5% (N = 200); and “Submission-Domination”, with 34.6% (N = 101) and 65.4% (N = 191), respectively.

In relation to gender, Table 3 shows the values of motivation in relation to this variable. Statistically significant variations exist (*p* < 0.050) in the “intrinsic motivation” dimension (*p* = 0.000) and in the “amotivation” dimension (*p* = 0.000), with no statistically significant differences in the “identified regulation” and “external regulation” dimensions, with women (intrinsic motivation M = 5.34; SD = 1.35; amotivation M = 1.62; SD = 0.92) having higher values than men in both dimensions.

Likewise, in the relationship established between motivation and social skills, Table 4 shows the relationship between the four established social skills and the four dimensions of the situational motivation scale. As shown, no statistically significant differences (*p* > 0.050) were found in attitude, activity and situational reaction. In the relationship between “intrinsic motivation” and “sociability” (*p* = 0.014), statistically significant differences were detected (*p* < 0.050), with the highest values in the sociable pole of the dichotomy (M = 5.19; SD = 1.38), a fact that is repeated in the relationship between the same and “identified regulation” (*p* = 0.012).

Table 4 reveals that there are statistically significant results (*p* < 0.050) between the students’ communicative ability and their “motivation” (*p* = 0.007), with the strongest values being found in reserved students (M = 2.08; SD = 1.29).

There were statistically significant results (*p* < 0.050) between students’ “intrinsic motivation” and “thinking” (*p* = 0.001), with the highest results for students who define themselves as optimistic (M = 5.31; SD = 1.31). There were also significant differences (*p* < 0.050) between this and the dimension “identified regulation” (*p* = 0.021), and between “thinking” and “motivation” (*p* = 0.019), with the strongest scores being found in “pessimistic” students (M = 2.02; SD = 1.18).

In relation to the ability to feel empathy or indifference, statistically significant differences (*p* < 0.050) were found between “intrinsic motivation” and this ability (*p* = 0.001), with the highest values being found in “empathetic” students (M = 5.20; SD = 1.36). Likewise, statistically significant differences (*p* < 0.050) were found between “amotivation” and this ability (*p* = 0.009), where the highest values were found in the “indifferent” students (M = 2.29; SD = 1.20). No significant results (*p* > 0.050) were found for “identified regulation” (*p* = 0.355) and “external regulation” (*p* = 0.458).

With regard to the relationship between motivation and the emotional reaction of the students, Table 4 shows the existence of statistically significant differences (*p* < 0.050) between “intrinsic motivation” and this (*p* = 0.000), with the highest values amongst the “assertive” participants (M = 5.30; SD = 1.31); and between “motivation” and “emotional reaction” (*p* = 0.007), with higher values being found in “aggressive” students (M = 2.11; SD = 1.27).

In relation to the last two social skills (social control and adaptability), as shown in the table, no significant statistical variation (*p* > 0.050) was found among the four dimensions and social control, nor between these and the adaptability of the students.

## 4. Discussion

The aim of this study was to carry out an analysis of the motivation of university students who are studying in the cross-border cities of Ceuta and Melilla, as cross-border contexts can have a significant impact on motivation. The unique cultural and linguistic environment of these cities can present both opportunities and challenges for students [24,38,40,41].

On the one hand, students who are exposed to different cultures and languages may develop a more open and diverse perspective, which may lead to higher levels of intrinsic motivation, even if this is more integrative than behavioural in nature [49]. Students who participate in activities that are new and exciting for them, such as learning about new cultures or languages, may experience a greater sense of personal fulfilment, which may contribute to an increase in their intrinsic motivation, with cultural enrichment as a motivational trigger [44,50].

On the other hand, students who are unfamiliar with the local culture and language may face challenges related to cultural adaptation and language proficiency, which may have an impact, leading to an increase in their extrinsic motivation or levels of amotivation [50]. Students who struggle to adapt to the local environment may feel isolated or disconnected from their peers, which may negatively affect their motivation to succeed academically, given the socio-cultural uprooting they experience [24,51].

This study reveals that the population in this case is predominantly female; that the predominant motivation is intrinsic; and that it is higher in women than in men [52,53,54]. These data contrast with other studies [43,55,56,57], in which the male population showed higher levels of motivation than the female population.

Overall, our findings indicate that intrinsic motivation is the predominant form of motivation among college students, with females scoring higher than males on average. We also observed a negative correlation between intrinsic motivation and amotivation, meaning that an increase in one is accompanied by a decrease in the other and vice versa [58,59].

We further discovered a direct correlation between student motivation and their social skills. Specifically, we found that student social competence is related to the regulation of intrinsic motivation and determination. This relationship is attributed to the internalization process of the motivational trajectory, which enables extroverted individuals to initiate and internalize extrinsic motivation, such as social approval. These findings are supported by previous research [1,21,60].

Another social skill that interacts with motivation, in this case with amotivation, is the ability to communicate, and we found results of motivation in students who define themselves as reserved; this may be caused by a negative evaluation of the result within the act of social communication on the part of the students, who find communication futile when they are the protagonists of it, in contrast to other results where students valued empathy and communicativeness as important and main skills [45,57,61]. This is similarly so for the pessimistic thinking student population. These data confirm that social skills are affected by personality, family and other aspects, observing an interrelation between social skills, motivation and self-regulation in university students in the southern United States [21,39,50,62].

In terms of empathetic attitude, empathetic people show greater intrinsic motivation than non-empathetic or indifferent people, similar to what has been obtained in other studies in which they pointed out motivation as a relevant factor in the development of empathy, related to the “individual stage of development linked to the human quality of each student”. All this is related to the initial motivation that prompted students to enrol in a university education [38,60,63].

Likewise, in terms of the social skill of assertiveness and its opposite, aggressiveness, the data reflect that assertiveness is related to high levels of intrinsic motivation, as well as other studies which point to assertive communication being directly related to high motivation and good leadership. In contrast, aggressive behaviour is related to high levels of motivation. This study, in line with other studies, concluded that there is a strong relationship between students’ motivation and aggressive behaviour, and that the relationship between them is inversely proportional, i.e. the higher the motivation, the lower the presence of aggressive behaviour in students.

Finally, analysing the social skill of assertiveness and its opposite skill, aggressiveness, the data obtained show that assertiveness is related to high levels of intrinsic motivation, which is related to high motivation and good leadership [64,65]. In contrast, aggressive behaviour is related to high levels of amotivation, and there is a strong relationship between motivation and aggressive behaviour in students, since an increase in one would lead to a decrease in the other and vice versa [66].

By understanding the factors that influence the motivation of university students in cross-border contexts, strategies can be designed to enhance their learning experience. This may include academic support programmes, tutoring, and cultural integration activities, among others [41,44,66,67,68,69].

## 5. Conclusions

This study shows that there is a larger female population in the degrees offered at the Ceuta and Melilla campuses (University of Granada).

The predominant motivation in the population is intrinsic, with higher figures in the female population; therefore, women are more strongly intrinsically self-motivated than is the case for men.

According to the research findings, there exists a significant correlation between a student’s social abilities and their level of motivation. Specifically, sociability, empathy and assertiveness have been linked to greater levels of intrinsic motivation when compared to other social skills. Conversely, communicativeness, pessimism and aggressiveness have been associated with reduced levels of intrinsic motivation and elevated levels of amotivation. These results indicate an inverse relationship between intrinsic motivation and amotivation. High scores in one area are linked to lower scores in the other and vice versa.

This relationship between motivation and social skills reflects the need to develop social skills training programmes for university students in Ceuta and Melilla in order to improve their level of motivation and social performance.

The main limitation of this study was the high level of absenteeism in some degree programmes. Therefore, and as a future line of research, it is proposed to extend the sample in those degree programmes in which a low participation rate is obtained.

Overall, the cross-border context of Ceuta and Melilla can have both positive and negative impacts on the motivation of university students. It is important that universities provide adequate support to ensure that university students can succeed academically and personally in these environments, as they can foster development for a future career that will take place in a culturally diverse environment.

## Figures and Tables

**Table 1 ijerph-20-05924-t001:** Socio-demographic data (gender, campus, religion and degree).

		Gender	N (%)	Total (% Total)			Gender	N (%)	Total (% Total)
Population		Male	97 (33.2%)	292 (100%)	Grade	Children’s education	Male	8 (2.7%)	36 (12.3%)
		Female	195 (66.8%)				Female	28 (9.6%)	
						Primary education	Male	16 (5.5%)	41 (14%)
Campus	Ceuta	Male	70 (24%)	229 (78.4%)			Female	25 (8.6%)	
		Female	159 (54.5%)			Social ed.	Male	7 (2.4%)	33 (11.3%)
	Melilla	Male	27 (9.2%)	63 (21.6%)			Female	26 (8.9%)	
		Female	36 (12.3%)			ADE	Male	24 (8.2%)	54 (18.5%)
							Female	30 (10.3%)	
Religion	Crist. Cathol.	Male	52 (17.8%)	163 (55.8%)		Primary education and CC. Act. Phys. and Sport	Male	15 (5.1%)	20 (6.8%)
		Female	111 (38%)				Female	5 (1.7%)	
	Musul.	Male	21 (7.2%)	75 (25.7%)		Nursing	Male	23 (7.9%)	101 (34.6%)
		Female	54 (18.5%)				Female	78 (26.7%)	
	Hindu	Male	1(0.3%)	1 (0.3%)		Business Administration and Law	Male	3 (1.0%)	6 (2.1%)
		Female	0 (0.0%)				Female	3 (1.0%)	
	Other	Male	23 (7.9%)	53 (18.2%)		RR. LL. and HR. HH.	Male	1 (0.3%)	1 (0.3%)
		Female	30 (10.3%)				Female	0 (0.0%)	

Note: Cathol. Christian/Catholic Religion; Musul. Muslim Religion; Early Childhood Education Degree in Early Childhood Education; Primary Education Degree in Primary Education; Social Education Degree in Social Education; Business Administration Degree in Business Administration and Management; Primary Education and Act. Act. Phys. and Sport Degree in Primary Education and Physical Activity and Sport Sciences; RR. LL and HR HH Degree in Labour Relations and Human Resources.

**Table 2 ijerph-20-05924-t002:** Socio-demographic data II (reason for enrolment and social skills).

Reason for Registration							
	Gender		Total		Gender		Total
	M	F			M	F	
Family expectations	4 (1.40%)	4 (1.40%)	8 (2.70%)	Opting for a better job category	7 (2.40%)	8 (2.70%)	15 (5.10%)
Preference	42 (14.40%)	21 (7.20%)	63 (21.60%)	Enhancing the curriculum	3 (1.00%)	1 (0.30%)	4 (1.40%)
Vocation	106 (36.30%)	39 (13.40%)	145 (49.70%)	To have more professional knowledge	2 (0.70%)	5 (1.70%)	7 (2.40%)
Have a qualification to work	6 (2.10%)	4 (1.40%)	10 (3.40%)	Lack of options	15 (5.10%)	9 (3.10%)	24 (8.20%)
Personal satisfaction	9 (3.10%)	4 (1.40%)	13 (4.50%)	Another	1 (0.30%)	2 (0.70%)	3 (1.00%)
**Social Skills**							
	**Gender**		**Total**		**Gender**		**Total**
	**M**	**F**			**M**	**F**	
Introversion	68 (23.6%)	28 (9.6%)	97 (33.2%)	Pessimism	73 (25%)	28 (9.6%)	101 (34.6%)
Extroversion	126 (43.2%)	69 (23.6%)	195 (66.8%)	Optimism	122 (41.8%)	69 (23.6%)	191 (65.4%)
Passivity	42 (14.4%)	22 (7.5%)	64 (21.9%)	Indifference	20 (6.8%)	12 (4.1%)	32 (11%)
Activity	153 (52.4%)	75 (25.7%)	228 (78.1%)	Empathy	175 (59.9%)	85 (29.1%)	260 (89%)
Reactivity	65 (22.3%)	27 (9.2%)	92 (31.5%)	Aggressiveness	43 (14.7%)	28 (9.6%)	71 (24.3%)
Proactivity	130 (44.5%)	70 (24%)	200 (68.5%)	Assertiveness	152 (52.1%)	69 (23.6%)	221 (75.7%)
Associateability	25 (8.6%)	15 (5.1%)	40 (13.7%)	Submission	73 (25%)	28 (9.6%)	101 (34.6%)
Sociability	170 (58.2%)	82 (28.1%)	252 (86.3%)	Domination	122 (41.8%)	69 (23.6%)	191 (65.4%)
Reserved	53 (18.2%)	28 (9.6%)	81 (27.7%)	Inflexibility	18 (6.2%)	9 (3.1%)	27 (9.2%)
Communicative	142 (48.6%)	69 (23.6%)	211 (72.3%)	Adaptability	177 (60.6%)	88 (30.1%)	265 (90.8%)

**Table 3 ijerph-20-05924-t003:** Relationship of motivation to gender.

Gender		Media	SD	F	*p*
**Intrinsic Motivation**	Female	5.34	1.35	16.852	0.000
	Male	4.64	1.45		
**Regulation Identified**	Female	6.20	0.94	3.37	0.067
	Male	5.99	0.88		
**External Regulation**	Female	3.42	1.43	1.769	0.185
	Male	3.66	1.46		
**Amotivation**	Female	1.62	0.92	18.804	0.000
	Male	2.19	1.31		

**Table 4 ijerph-20-05924-t004:** Relationship between motivation and social skills.

Attitude		M	SD	F	*p*	Activity	M	SD	F	*p*	Communication		M	SD	F	*p*
MY	INT	4.97	1.44	1.548	0.214	PAS	4.86	1.51	2.546	0.112	RES		4.88	1.49	2.885	0.090
	EXT	5.18	1.40			ACT	5.18	1.38			COM		5.20	1.38		
RI	INT	6.08	0.91	0.344	0.558	PAS	6.06	0.91	0.404	0.526	RES		6.01	0.96	1.558	0.213
	EXT	6.15	0.94			ACT	6.14	0.93			COM		6.17	0.91		
RE	INT	3.59	1.38	0.525	0.469	PAS	3.53	1.39	0.036	0.849	RES		3.55	1.48	0.143	0.705
	EXT	3.46	1.47			ACT	3.49	1.45			COM		3.48	1.43		
AM	INT	1.87	1.09	0.490	0.485	PAS	1.98	1.16	1.985	0.160	RES		2.08	1.29	7.266	0.007
	EXT	1.78	1.10			ACT	1.76	1.07			COM		1.70	0.99		
**Situational Reaction**		**M**	**SD**	**F**	** *p* **	**Sociability**	**M**	**SD**	**F**	** *p* **	**Thinking**		**M**	**SD**	**F**	** *p* **
MY	RE	5	1.49	0.849	0.357	ASO	4.6	1.56	6.150	0.014	PES		4.72	1.53	11.934	0.001
	PRO	5.16	1.39			SOC	5.19	1.38			OPT		5.31	1.31		
RI	RE	6.09	0.93	0.165	0.685	ASO	5.79	1.04	6.369	0.012	PES		5.95	1.03	5.418	0.021
	PRO	6.14	0.92			SOC	6.18	0.89			OPT		6.22	0.85		
RE	RE	3.55	1.55	0.166	0.684	ASO	3.37	1.31	0.352	0.553	PES		3.62	1.55	0.99	0.320
	PRO	3.48	1.39			SOC	3.52	1.46			OPT		3.44	1.38		
AM	RE	1.90	1.17	0.840	0.360	ASO	2.10	1.22	3.406	0.066	PES		2.02	1.18	5.6	0.019
	PRO	1.77	1.06			SOC	1.76	1.07			OPT		1.70	1.04		
**Capacity**		**M**	**SD**	**F**	** *p* **	**Emotional Reaction**	**M**	**SD**	**F**	** *p* **	**Adaptability**		**M**	**SD**	**F**	** *p* **
MY	IND	4.36	1.69	10.44	0.001	AGR	4.54	1.59	16.124	0.000	INF		4.72	1.37	2.261	0.134
	EMP	5.20	1.36			ASE	5.30	1.31			ADA		5.15	1.42		
RI	IND	5.98	0.84	0.859	0.355	AGR	5.96	1.00	3.048	0.082	INF		6.12	0.86	0.002	0.966
	EMP	6.14	0.93			ASE	6.18	0.89			ADA		6.13	0.93		
RE	IND	3.68	1.49	0.552	0.458	AGR	3.49	1.57	0.003	0.958	INF		3.51	1.57	0.001	0.975
	EMP	3.48	1.44			ASE	3.50	1.40			ADA		3.5	1.43		
AM	IND	2.29	1.20	6.989	0.009	AGR	2.11	1.27	7.291	0.007	INF		2	1.12	0.892	0.346
	EMP	1.75	1.07			ASE	1.71	1.02			ADA		1.79	1.09		
**Control**		**M**	**SD**	**F**	** *p* **	**Control**		**M**	**SD**	**F**		** *p* **				
MY	SUM	5.18	1.3626	0.324	0.569	RE	SUM	3.50	1.356	0.001		0.972				
	DOM	5.08	1.4529				DOM	3.50	1.48888							
RI	SUM	6.10	0.95602	0.122	0.727	AM	SUM	1.76	0.99271	0.289		0.591				
	DOM	6.14	0.91172				DOM	1.83	1.15093							

Note: IM: intrinsic motivation; IR: identified regulation; ER: external regulation; AM: amotivation; INT: introvert; EXT: extrovert; PAS: passive; ACT: active; RES: reserved; COM: communicative; RE: reactive; PRO: proactive; ASO: asocial; SOC: sociable; OPT: optimistic; PES: pessimistic; IND: indifferent; EMP: empathetic; AGR: aggressive; ASE: assertive; INF: inflexible; ADA: adaptive; SUM: submissive; DOM: dominant. M: media. F: frequency. SD: standard deviation. *p*: *p* value.

## Data Availability

The data, search strategies, results on each database and main characteristics of the studies included in the systematic review used are provided in annexes and remain in the custody of the author of the correspondence. Anyone interested can request them from the author of the correspondence: jcortesmartin@ugr.es.

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
