# Peer review of "Analysis of University Student Motivation in Cross-Border Contexts"

_ijerph, 2023, doi:10.3390/ijerph20115924_

Round 1
Reviewer 1 Report
First of all, I would like to express my appreciation to the authors for choosing such a debated topic. I formulate the following comments with the main purpose of improving the work carried out so far.
- The introduction section must be improved by adding more recent publications and by making it more specific for the target population (higher-education students).
- The in-text citation should follow the journal's regulations.
- The study's objectives and employed instruments are insufficiently or not clearly described.
- The authors should be consistent in reporting the results throughout the text.
- The results and the resulting tables should be reported following the APA standards.
- The authors should explain why choosing the Anova method instead of more advanced statistical techniques.
Author Response
Dear reviewer,
Thank you very much for your recommendations, which have been integrated into the article and have improved it considerably. All changes are highlighted in the new version of the article.
Kind regards.

Reviewer 2 Report
The article presented in the Journal presents an interesting approach to a particular phenomenon. The theoretical approach is clear, detailed and coherent, it manages to place the reader in the question of study. Important the classification of the motivations that are provided, however are not exploited in the methodological instrument, results, discussion and conclusions. This would be a first adjustment factor in the development of the article.
The methodology is clear, but it would go a long way in detailing the component segments of the instrument, their connection with the research objectives or questions and the categories of analysis used. Detailing the methodological elements strengthens the work and gives consistency to the results.
Working in line with the categories of analysis and connecting with the motivations set out in the theoretical framework would help to exploit the results you have. It is important to go beyond what is evident in the instrument. The triangulation of factors can give greater significance to what is obtained with the sample.
It is essential that the authors develop a discussion with the authors who posed in the theoretical framework. Complementing, refuting or affirming the authors' proposals will be a fundamental contribution of this work. This proposed exercise will connect the conclusions and give relevance to them.
Author Response

(The authors gave the same response as above.)

Reviewer 3 Report
Analysis of university student motivation in cross-border contexts
This paper presents a quantitative study exploring students’ motivation in relation to a number of variable such as sociability, communication, thinking (optimistic or pessimistic), empathy and self-confidence, while gender difference are considered as well.
In general, the paper is quite interesting. However, in my opinion, there are some drawbacks and serious concerns regarding methodological issues and the presentation of the results.
Comments
-
The abstract needs to be rewritten and include some information about the methodology followed and clearly the finding as well.
-
Consider to move the methods of data analysis to “procedures” section.
-
Provide reliability measures of the motivational dimensions
-
Line 170: please report the variability of age with standard deviation (not only with max-min values).
-
Use consistently through the text, the word Table, with “T” not small latter ‘t”.
-
Table 2: explain the symbols M and H (is it F?)
-
‘Social skills’ are not elaborated satisfactorily in the introduction section. What is the level of measurement for these variables? Are they categorical variables? It should be mentioned in the “instrument” section. Introversion, for example is categorical (yes/no), not a Likert type interval scale variable?
-
Table 3: It should be explicated that the results form a one way ANOVA. T-test, however, is preferable for the differences between two groups.
-
Table 3: Are “Media’ and ‘DT’ the “mean’ and SD (standard deviation)?
-
Figure 1 is actually a Table (Table 4?).
-
Figure 1: again, are “M’ and ‘DT’ the “mean’ and SD (standard deviation)?
-
Figure 1: It should be revised. There are a lot of inconsistences regarding abbreviations and symbols. For example: what is MY, also check IR RI, ER or RE (are they the same). Please check and correct.
-
Figure 1: There are many insignificant differences. I think, it is worth presenting only the statistically significant results. It will make the Figure/Table more readable.
-
Figure 1 present the results of ANOVA test using categories from Table 2.
Many of these categories include small number of cases.
It is advisable that this descriptive information should be included in this ANOVA table. To avoid the Table from becoming complicated, exclude the insignificant difference and present the Table with more information, e.g. the values of η2 and effect sizes.
-
Provide a section mentioning the limitation of the study.
Author Response
Dear reviewer,
Thank you very much for your recommendations, which have been integrated into the article and have improved it considerably. All changes are highlighted in the new version of the article. We have included figure 1 as table 4 following your suggestion for a change.
Kind regards.

Reviewer 4 Report
Thank you for giving me the opportunity to review this paper. Based on the questionnaire data of 292 college students, this paper explored the motivational characteristics of college students and their gender differences. The paper has the following problems:
1. The definition of "cross-border contexts" in the title is not clear.
2. Does the sample selection follow the random sampling procedure? Is it representative of college students?
3. The paper lacked necessary explanation for the reliability and validity of the scale.
4. The statistical analysis in this paper only based on the correlation analysis of bivariables, which is difficult to clarify the causal relationships between different variables.
Author Response
Dear reviewer,
Thank you very much for your recommendations, which have been integrated into the article and have improved it considerably. All changes are highlighted in the new version of the article.

Round 2
Reviewer 1 Report
accept in the present form
Reviewer 3 Report
Τhe authors have addressed my comments, so I endorse the publication of the manuscript.
A language proofreading prior to the publication is needed.
Reviewer 4 Report
The authors have revised the paper according to the previous suggestions. The present paper has basically met the publication requirements.